# LEARNING TO REASON WITH RELATIONAL ABSTRACTIONS

## ABSTRACT

Large language models have recently shown promising progress in mathematical reasoning when fine-tuned with human-generated sequences walking through a sequence of solution steps. However, the solution sequences are not formally structured and the resulting model-generated sequences may not reflect the kind of systematic reasoning we might expect an expert human to produce. In this paper, we study how to build stronger reasoning capability in language models using the idea of relational abstractions. We introduce new types of sequences that more explicitly provide an abstract characterization of the transitions through intermediate solution steps to the goal state. We find that models that are supplied with such sequences as prompts can solve tasks with a significantly higher accuracy, and models that are trained to produce such sequences solve problems better than those that are trained with previously used human-generated sequences and other baselines. Our work thus takes several steps toward elucidating and improving how language models perform on tasks requiring multi-step mathematical reasoning.

## 1 INTRODUCTION

Deep learning has had tremendous success in a wide range of domains, such as vision (He et al., 2016), language (Brown et al., 2020), and playing games at superhuman levels (Mnih et al., 2015; Silver et al., 2016; Vinyals et al., 2019). Yet despite these accomplishments, these systems remain limited in their formal and mathematical reasoning abilities (Saxton et al., 2019; Cobbe et al., 2021; Hendrycks et al., 2021). Although there have be recent impressive gains Lewkowycz et al. (2022), the models remain challenged to succeed at harder problems.

Recent work suggest that neural networks, like humans, benefit from relying on a chain of reasoning steps rather than attempting to produce the final output as a direct mapping from the problem prompt (Recchia, 2021; Nye et al., 2021; Hendrycks et al., 2021; Cobbe et al., 2021; Lewkowycz et al., 2022). These works rely entirely on naturalistic data and manipulations, in the sense that problems and their step-wise solutions are taken as they are found in existing sources, or human annotators are asked to produce a sequence of solution steps using numbers interspersed with natural language. However, while naturalistic sentences are certainly how we often communicate our solutions to each other informally, we argue that formal and mathematical reasoning depends on identifying and exploiting the set of abstract relationships that underlies the details of the problem at hand. Even in settings where the focus is on the step-wise manipulation of quantities to obtain valid practical results, a set of abstract relationships underlies the sequence of operations.

We build on this intuition by exploring the possibility that, if a problem-solver can formulate the problem under consideration at an abstract level, this will be conducive to finding the correct sequence of more specific arithmetic operations. However, to our knowledge, no math dataset currently exists that utilizes natural language and also isolates key reasoning components such as entities and their relations, i.e. there is no way to train the model to convert natural language inputs into these core elements. We address this gap by proposing a new dataset, GSM8K-R, by expanding on the GSM8K dataset (Cobbe et al., 2021), a dataset containing grade-school level math word problems, with human annotations that highlight the relational abstractions that are central to mathematical reasoning. We also introduce a new synthetic task, called the unit conversion (UC) task, in which the abstract relational problem is reduced to its essence that enables controlled analyses without the complications that arise from naturalistic datasets.

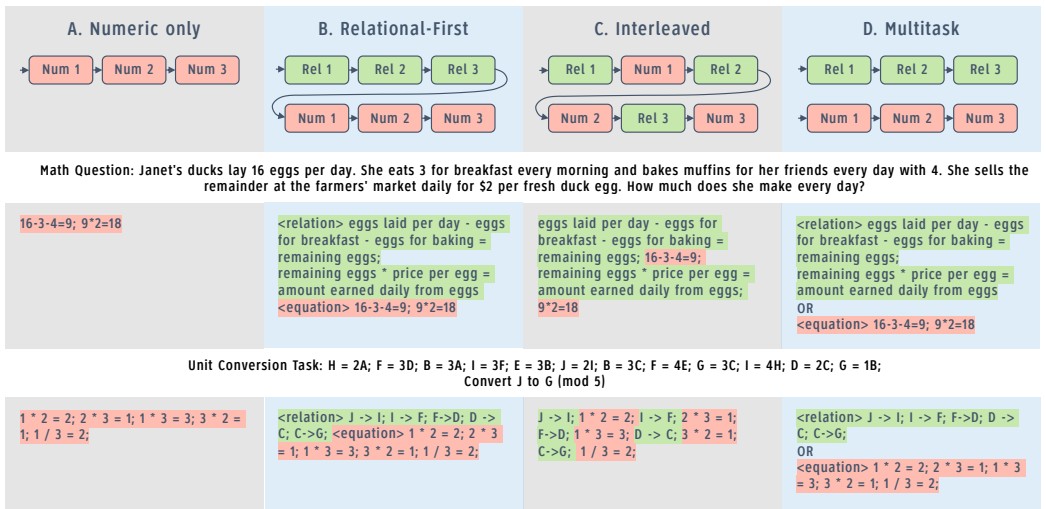

Figure 1: We explore abstract relational reasoning by partitioning the reasoning process into the *abstract relational* and the *numeric* part, and compare four different possibilities: **Numeric only (NN)**: Only numeric steps are provided without any relational tokens; **Relational-first: (RRNN)** The abstract relational parts are stated before the numeric; **Interleaved: (RNRN)**: relational then numeric parts occur in alternating sequence; and **Multitask: (RR|NN)**: The network learns to produce either the abstract relational or the numeric sequence to a task prompt, then prompted for the numeric sequence at test time.

At their core, both tasks involve reasoning about how different quantities relate to each other, and formulating appropriate arithmetic equations to perform the corresponding numerical computations. We can decompose each step of the solution into abstract relational reasoning and arithmetic expressions, which can then be used to recompose the solution sequence in different forms.

We summarize our main contributions as follows:

- We decompose the problem solving process into identifying the relevant abstract relationships and performing the corresponding arithmetic manipulations.
- We present a new dataset called GSM8K-R that adds relational abstraction annotations to the original GSM8K dataset (Cobbe et al., 2021) (to be released with the paper).
- We introduce the new synthetic task Unit Conversion task that brings out the importance of engaging with the relational abstractions, even in smaller transformer models.
- We find that teaching models to identify the relevant abstract relationships on trained problems can lead to substantial performance gains at test, and identify several factors affecting this outcome.
- We find that identifying the crucial abstract relationships remains a challenge, and that providing the relational abstraction at test time can produce drastic gains.

Taken together, we believe these findings highlight the importance of identifying the relevant abstract relations to enable correct formal and mathematical reasoning. In the discussion, we consider next steps that may allow the development of artificial systems that capture this ability.

## 2 INCORPORATING RELATIONAL ABSTRACTION

In this section, we describe our framework of incorporating relational abstractions into mathematical reasoning. We begin with the notion that mathematical problem solving involves determining the values of unknown quantities from known quantities, where a quantity is a numerical attribute of an item or set, such as the price of an item or the number of items in the set. Quantities can be derived from other quantities relying on rules that apply to quantities of relevant types. For example, as in the problem shown in Table 1, the amount earned from selling some number of items (in this case, eggs) is equal to the product of the number of items sold times the price per item.

In general, mathematical problem solving requires several operations on given quantities to obtain a final answer – a specified target or goal quantity. In the problem in Table 1, we are given the number of eggs Janet's ducks lay each day, eggs eaten for breakfast, eggs used in baking, and we are told that she sells the remainder for a specified price per egg. To solve for how much money she makes, we must first determine the remainder by subtracting the number of eggs eaten and the number of eggs used in baking from the number laid, and then determine the amount earned by multiplying the remaining number of eggs times the price per egg.

This exemplifies what we call the abstract relational plan: a plan outlining the reasoning process without invoking any numbers. Here, "eggs laid", "eggs eaten", "eggs used in baking", "remaining eggs" and "price per egg" are quantities needed to reach the target quantity. The abstract relational plan specifies the steps that must be applied to the given quantities to reach the relevant intermediate quantities, and then applied to these quantities to reach the final answer. What makes a plan abstract is that it omits specific information – that is, the specific quantities involved – and connects items through how they relate to each other at a more general or abstract level. What makes it relational is that it specifies which entities are relevant to each other in the problem. An abstract relational plan formulates the problem as a graph of interconnected abstract entities, whose specific values could be replaced by others without changing the set of relationships.

The problems found in the GSM8K dataset can all be seen as solvable by extracting the correct abstract relational plan from the verbal statement of the problem and then applying the plan to obtain the numeric value of the target quantities given the values of the given quantities. The challenge here is that GSM8K, and other math datasets like it, consists entirely of natural language data that makes it difficult to systematically extract the relevant entities and their relations. We address this issue through our human-annotated dataset GSM8K-R that provides the ground truth labels to train the model with, and we explore several instructional forms that utilize these annotations.

Figure 1 enumerates a few possibilities for how we can incorporate abstract relational reasoning into the training and testing of a decoder-only transformer of the kind used in the GPT model series. We first decompose a solution sequence into an an abstract relational plan, consisting of a sequence of abstract relational expressions as described above and a sequence of arithmetic expressions involving only numbers and basic arithmetic operations. We can then train and test the models using conditions of the following four types: *numeric-only* (NN) uses only the $n$ arithmetic sequences, and serves as our baseline. In *relational-then-numeric*, (RRNN) the relational expressions are presented before numeric ones. This represents the strategy of generating a high-level relational plan first, and then implementing the plan by performing the relevant arithmetic operations. The *interleaved* format (RNRN) alternates between the abstract relational expressions and the arithmetic expressions, so that each arithmetic expression is accompanied by the relevant abstract relational expression. Lastly, in the *multitask* approach (RR| NN), the model is prompted to output the sequence of *either* the relational *or* the numeric expressions, but not both. This may allow the model to learn to represent the problem at the abstract level and exploit such representations even when it is only producing the numerical expressions. This approach tests the claim that additional auxiliary language tokens effectively function as regularizers or learning tools that can be discarded at test time and may even suppress performance if included (Mu et al., 2020; Lampinen et al., 2022; Hendrycks et al., 2021). Moreover, learning and generating the two sequences separately has the added advantage of generating shorter sequences at test time, just like numeric-only. In this paper, we examine which type of relational abstraction brings the best reasoning capability in each of our two task settings.

## 3 RELATED WORK

Although computational models of mathematical reasoning have been proposed for over half a century (Bobrow, 1964), application of neural network models began much more recently using recurrent networks for sequence-to-sequence prediction (Wang et al., 2017). Shortly after their introduction in Vaswani et al. (2017), Saxton et al. (2019) found that transformers-based models outperformed other architectures when trained to generate the answer directly from the problem statement. Many researchers have explored enhancing model performance by fine-tuning to produce intermediate equations or programs (Shi et al., 2015; Upadhyay & Chang, 2015; Amini et al., 2019; Miao et al., 2020; Drori et al., 2021). Recent advances rely on large transformer-based language models (Brown et al., 2020; Thoppilan et al., 2022; Chowdhery et al., 2022; Lewkowycz et al., 2022) and/or datasets involving full step-by-step solutions in natural language (Ling et al., 2017; Hendrycks et al., 2021; Welleck et al., 2021; Cobbe et al., 2021; Drori et al., 2021).

Interestingly, prompting large language models such as GPT-3 to generate chains of thought with just a few examples at test time can enhance performance considerably (Wei et al., 2022), indicating that the models may already have the ability to engage in a step by step reasoning process, in part because such a process is exemplified in their training. Many recent works use multiple samples from a model, either using a verifier trained on model-generated responses to re-rank candidate sequences Cobbe et al. (2021) or relying on a majority voting scheme (Wang et al., 2022). The strongest results

overall to date (Lewkowycz et al., 2022) use a very large transformer based language model, fine-tuned on scientific and mathematical text, provided with a chain of thought prompt, and assessed using majority voting. However, these models still only achieve modest scores on harder problems, consistent with the view Hendrycks et al. (2021) that simply scaling up the model size is an intractable strategy for solving mathematics problems of higher difficulty, even with the added benefit of chain-of-thought prompting, verifiers, or majority voting.

Common across these existing works is the use of human-generated solution sequences. In our work, we introduce our GSM8K-R dataset to explicitly contrast performance on different types of solution sequences and explore how explicit focus on generating a structured abstract relational plan can improve learning, an analysis that would not be possible with existing datasets. We also introduce the unit conversion (UC) task, a completely synthetic task domain to complement our exploration of solving problems expressed in natural language. This parallels the approach of Gontier et al. (2020), with a crucial difference. These authors investigated logical reasoning over a fixed database of specific relational facts, training models to produce an inferable relation to a probe question, and found only small advantages of a plan sequence compared to generating the answer directly. In contrast, our UC task affords separating the abstract relational plan from the specific numerical computations. This allows us to demonstrate a striking advantage from learning to produce the abstract relational sequence rather than just the necessary numerical expressions.

## 4 EXPERIMENTS

We use two tasks to explore the possible benefits or relational abstractions: a set of natural language math problems from the Grade School Math 8K (GSM8K) dataset (Cobbe et al., 2021), and an abstract unit conversion task (UC) in which the model must determine how the number of units of one type corresponds to a specified number of units of another type. Both tasks contain quantities and relations that can be represented by a graph, and involve formulating and solving a series of numerical equations. However, the two tasks pose different challenges, allow different approaches to model training, and afford different comparison conditions and analyses.

The GSM8K dataset consists of realistic word problems requiring a broad understanding of mathematical concepts and their application to grade school math problems. The dataset includes human-generated mixed expressions that usually step through the problems in a linear order corresponding to the problem statement in a fairly small number of solution steps. Because these are word problems, they challenge the model's natural language understanding and general world knowledge (such as the fact that a dozen consists of 12 items, or that the number of eggs increases when it is laid by a chicken but decreases when it is used in baking cookies). We present our GSM8K-R dataset by building on the GSM8K dataset, adding human annotations that extract the core components of the reasoning process, namely the entities, quantities, and the arithmetic operations that define the entities' relations. In this setting we fine-tune pre-trained language models and compare our proposed conditions to the natural language based comparison conditions provided with the data set.

The unit conversion task avoids the natural language understanding and world knowledge issues by presenting conversion rules in a simple symbolic form. This allows us to present problems requiring the use of a larger number of specified relationships that are presented to the model in a random order and requiring longer sequences of solution steps. In this setting we use smaller scale models that we are able to train end-to-end, allowing us consider several additional variations of the training regime and to analyze the model's step-by-step performance more straightforwardly. Together our two tasks offer both a rich, naturalistic environment with empirical results for broader applicability and a systematic, synthetic environment that reduces mathematical reasoning to its most abstract form, bringing out the advantage of relational abstractions more clearly.

Table 1 presents key results from the four conditions illustrated in Figure 1. In both the GSM8K-R and UC tasks, the models perform very poorly after fine tuning to generate the answer directly from a problem statement (25% correct is the chance level on the UC task), and training on numeric sequences produces some improvement for GSM8K-R but only a hint of a gain over chance level for UC. The multitask condition produces slight gains for but models, but the real big gains are observed when the models have been trained to produce relational sequences either before or alternating with the numerical sequences. For GSM8K-R, the benefit only appears when the relational plan is included in the prompt at test time. In the UC setting, we also see big gains when the model produces the relational sequence for itself, and we also see that this advantage comes only on trials where the

| | Model, training regime, and dataset | | | |
| --- | --- | --- | --- | --- |
| | GPT2-XL, fine-tuned on GSM8K | | Simple transformer trained from scratch on unit conversion dataset | |
| Type of steps in training | Problem prompt only | Problem & relational plan prompt | Problem prompt only | Problem & relational plan prompt |
| Answer only baseline | 4.93 | - | 24.7 | - |
| Numeric only (NN) | 22.97 | - | 25.9 | - |
| Multitask (RR\|NN) | 28.05 | - | 29.8 | - |
| Relational First (RRNN) | 19.48 | 64.59 | 71.1 W/ correct plan: 96.8 W/ incorrect plan: 21.0 Plan accuracy: 66.2 | 96.7 |
| Interleaved (RNRN) | 22.97 | 66.26 | 85.8 W/ correct plan: 99.9 W/ incorrect plan: 20.3 Plan accuracy: 82.3 | 99.9 |

Table 1: Key results demonstrating the key findings from the parallel conditions of our two experiments. Fuller definition of the conditions are given in the caption for Figure 1.

model produces the relational sequence correctly. Indeed, either when the model produces the relational sequence correctly itself or when prompted with the correct relational sequence, performance is at near-ceiling levels. In the next sections we describe the two data sets and experiments in more detail, along with further many findings from many additional comparison conditions.

### 4.1 TASK 1: SOLVING GRADE SCHOOL MATH PROBLEMS

We first evaluate our framework on more realistic problems posed using natural language in the GSM8K-R dataset, which contains around 7.7K training question and 1.3K test questions from the original GSM8K dataset with additional human annotated solutions, all in the form of the English language. An example of the problem and its solution can be found in the first two rows of Table 2. The original dataset contains the following possible solution formats:

- The *original solution* format was used in the original paper. It provides solution steps in natural language annotated with executable equations. It is similar to our interleaved approach in that the target unit of each step often appears at the end of the sentence (e.g. Janet sells 16-3-4 *eggs* a day).

- The *equation-only* format contains the numerical equations without any use of natural language to reference any objects or units.

- The *socratic* version contains questions that ask for intermediate answers, which we can prepend before each step of the original solution (*socratic + solution*) or of the equation-only format (*socratic + equation*). The questions are in the GSM8K dataset, but prior work did not use them.

In addition to these formats, we introduce the *relation + equation* format that features relational abstractions. The input arguments and the types of transition functions are specified in addition to the output quantity. For example, "amount earned" is the step output, and "number of eggs multiplied by price per egg" is the relational statement needed to compute the output. Since the original dataset only contains language solutions without any additional labels, we asked human participants to annotate the entire GSM8K dataset so that each solution step would be paired with an abstract relation. We include our labeling task instructions in the Appendix C. Both the socratic and relation formats contain pairs consisting of an auxiliary sequence and a solution sequence. Following the setup outlined in Section 2, we either place the auxiliary sequence first or interleave it with the numerical expressions, which we refer to as *aux-first* and *interleaved* respectively in our results. We also include a *multitask* variant of our *relation* format. Here, during training, the model is prompted to generate relational sequences on 1/2 of the training batches, and numeric sequences on the other half, then prompted at teste time to generate numeric sequences.

**Implementation.** Following Cobbe et al. (2021) we use pretrained GPT2-M and GPT2-XL models (Radford et al., 2019), first fine-tuning the model on the question & answer sequences for 40

| Problem | Janet's ducks lay 16 eggs per day. She eats three for breakfast every morning and bakes muffins for her friends every day with four. She sells the remainder at the farmers' market daily for $2 per fresh duck egg. How much in dollars does she make every day at the farmers' market? |
|---|---|
| **Natural language** | |
| Original solution | (1) Janet sells 16 - 3 - 4 = ≪16-3-4=9≫ 9 duck eggs a day. 
 (2) She makes 9 * 2 = ≪9*2=18≫18 every day at the farmer's market. |
| **Numeric only** | |
| Equation only | (1) ≪16-3-4=9≫ 
 (2) ≪9*2=18≫ |
| **Socratic prompts** | |
| Socratic + solution | (1) How many eggs does Janet sell? Janet sells 16 - 3 - 4 = ≪16-3-4=9≫ 9 duck eggs a day. 
 (2) How much does Janet make at the farmers' market? She makes 9 * 2 = ≪9*2=18≫18 every day at the farmer's market. |
| Socratic + equation | (1) How many eggs does Janet sell? ≪16-3-4=9≫ 
 (2) How much does Janet make at the farmers' market? ≪9*2=18≫ |
| **Relational + numeric** | |
| Relation + equation | (1) eggs laid per day - eggs for breakfast - eggs for baking = remaining eggs ≪16-3-4=9≫ 
 (2) remaining eggs * price per egg = amount earned daily from eggs ≪9*2=18≫ |

Table 2: GSM math dataset sample problem and variants of solution sequence format.

epochs with the AdamW optimizer (Loshchilov & Hutter, 2019) and learning rate 1e-5. During testing, We generate output sequences and use the calculator to evaluate the equations as in Cobbe et al. (2021). We primarily use greedy decoding, but sometimes use a verifier following Cobbe et al. (2021) (see Appendix A.1 for details). When conditioning with ground-truth auxiliary prompts, we do not use verification as the same output samples are generated multiple times.

**Results.** Table 3 shows the main results using GPT2-M and -XL with greedy decoding. The larger language model achieves better performance across the board, though the margin varies with other factors. Note that our numbers are obtained using GPT-2, which is about $100\times$ smaller than GPT-3 in terms of parameter count, so lower accuracy is to be expected. Compared to the *answer-only* baseline, in which the intermediate steps are omitted, all of the multi-step approaches offer an improvement. Equation-only outperforms the original solution format (22.97% vs. 17.44%), which contains both numbers and text, and this advantage generally holds in other matched comparisons. When the model is fine-tuned with auxiliary sequences (socratic or relation sequences) paired with solution sequences (either the original GSM8K solution or our numeric equation sequences), we see generally worse performance when the model must generate both types of sequences compared to the numerical only cases. However, the sequences the model is fine-tuned with are quite long, and performance generally degrades as sequence length increases. Indeed, we find that accuracy generally decreases with increasing solution steps and answer length, and the equation only format suffers the most obvious degradation (see Appendix A.2 for details).

Our multitask regime avoids this difficulty. We see that multitask training leads to substantially improved performance in the larger GPT2-XL model (28.05% correct compared to the baseline of 22.97%, a 22% relative improvement). This finding shows clearly that training to reason relationally can improve test-time performance, even when at test-time the model is only generating numerical sequences. Relation + equation (interleaved) achieves better results than equation-only (29.49% vs. 24.79%), and is almost on par with multitask (29.49% vs. 30.17%) when using 20 samples and the external verifier. We find that verification is less helpful when the output format is purely numeric, such as in the multitask and equation only formats.

As noted previously, model accuracy improves significantly when models trained with auxiliary and solution sequences are prompted at test time with the ground-truth auxiliary sequence. Strikingly, prompting with ground-truth relational sequences triples the accuracy compared to the equation-only model (66.26% vs. 22.97%). Moreover, our relational sequences are far better prompts than the GSM8K socratic questions (66.26% vs. 36.92%), suggesting that with a good abstract relational plan, language models can solve the math questions much more easily. These results also indicate that the challenge the models face lies primarily in constructing the correct relational plan.

All else being equal, generating the full relational sequence first as an overall plan is nearly always slightly worse than interleaving relational and equation sequences, and this general pattern holds

| Method | GPT2-M (345M) | GPT2-XL (1.5B) | GPT2-XL (1.5B) + Verifier (345M) |
|---|---|---|---|
| Baseline without sequence generation | | | |
| Answer only | 3.56 | 4.93 | - |
| Solution sequences only | | | |
| Original Solution | 10.69 | 17.44 | 23.35 |
| Our Equation Only (NN) | **15.32** | **22.97** | **24.97** |
| Auxiliary and solution sequences: Model generates both | | | |
| Socratic + Soln. (aux-first) | 10.01 | 13.95 | - |
| Socratic + Soln. (interleaved) | 9.93 | 17.51 | - |
| Socratic + Eqn. (aux-first) | 13.27 | 19.03 | 23.35 |
| Socratic + Eqn. (interleaved) | 15.16 | 21.00 | 25.85 |
| Relation + Eqn. (aux-first) (RRNN) | 12.59 | 19.48 | 25.55 |
| Relation + Eqn. (interleaved) (RNRN) | 13.19 | 22.97 | 29.49 |
| Auxiliary and solution sequences: Trained to generate either, prompted for numeric at test | | | |
| Relation + Eqn. (multitask) (RN\|RN) | **15.62** | **28.05** | **30.17** |
| Auxiliary and solution sequences: Prompt with auxiliary, model generates solution sequence | | | |
| Socratic + Soln. (aux-first) | 17.46 | 26.23 | - |
| Socratic + Soln. (interleaved) | 17.89 | 28.89 | - |
| Socratic + Eqn. (aux-first) | 20.47 | 35.56 | - |
| Socratic + Eqn. (interleaved) | 27.82 | 36.92 | - |
| Relation + Eqn. (aux-first) | 54.59 | 64.59 | - |
| Relation + Eqn. (interleaved) | **58.53** | **66.26** | - |

Table 3: GSM-8K Finetuning Top-1 Test Solve Accuracy (%). Labels NN, RRNN, RNRN, and RR|NN designate conditions also shown in Table 1

throughout our results in Tables 3 and 7. The fact that this pattern continues when the relational sequences are provided as prompts suggests that proximity between the corresponding relational and numerical reasoning components helps the model retrieve the correct numeric information.

## 4.2 TASK 2: UNIT CONVERSION

The unit conversion task takes as input a given quantity and unit, then requires finding the equivalent quantity in another unit based on a set of conversion rules that are provided in the prompt (see Table 4). Problems of this type correspond abstractly to a subset of the problem types encountered in GSM8K. The conversion rules are presented in random order, and can collectively be viewed as edges of a graph. Although conversions are bidirectional, only one direction is specified directly in the prompt for each rule so that solving the task is equivalent to finding a path from the source node to the destination node while performing the corresponding multiplication (forward) or division (backward) operations when traversing each edge. This task offers a second context, using totally synthetic problems that eliminate any world knowledge and linguistic uncertainties that the GSM8K problems present, in which to explore the role of teaching the model to identify the abstract sequence of unit conversion steps rather than just step through the required sequence of numeric conversions. In this task setting, we find a very clear advantage from providing and training models to produce relational, as well as numeric, sequences compared to producing numbers alone.

The task (Table 4) is presented as a sequence completion task using the graph description and the conversion instruction as the task prompt. In addition to an answer-only baseline, we train the model to produce solution sequences. There are eight single-task conditions, using four sequence types each with or without an initial *relational-plan* specifying the sequence of units to traverse before producing the sequence containing numeric calculations. The four sequence types are *numeric-only*, containing only the numerical expressions, and three *interleaved* relational and numeric sequence types: *units-then-numbers* gives the source and destination units of the traversing edge followed by the numerical expression; *numbers-then-units* gives the numerical expression, followed by the source and destination units; *integrated* states the source quantity and unit, then the remainder of the numerical expression, followed by the destination unit.

As in the previous task, we also test each model's capacity to execute a provided correct relational plan by including the ground-truth plan as part of the given prompt for relational plan models. Lastly, as in the GSM8K experiments, we also consider the multitask approach in four more conditions, in which the network is prompted to generate either the relational plan or one of the four types of sequences. The subset of the full set of these conditions corresponding to the NN, RRNN, RNRN and RR|NN conditions as defined in Figure 1 are flagged in Table 5.

| Task Prompt | Relational Plan (Optional) | Sequence Types | | | |
|---|---|---|---|---|---|
| | | **Numeric Only** | **Interleaved** | | |
| | | | Units Then Numbers | Numbers Then Units | Integrated |
| graph $H \to 2$ A   $F \to 3$ D $B \to 3$ A   $I \to 3$ F $E \to 3$ B   $J \to 2$ I $B \to 3$ C   $F \to 4$ E $G \to 3$ C   $I \to 4$ H $D \to 2$ C   $G \to 1$ B convert 1 J to G | relations $J \to I \to F \to$ $D \to C \to G$ | steps $1 * 2 \to 2$ $2 * 3 \to 1$ $1 * 3 \to 3$ $3 * 2 \to 1$ $1 / 3 \to 2$  2 G  | steps J I $1 * 2 \to 2$ I F $2 * 3 \to 1$ F D $1 * 3 \to 3$ D C $3 * 2 \to 1$ C G $1 / 3 \to 2$  2 G  | steps $1 * 2 \to 2$ J I $2 * 3 \to 1$ I F $1 * 3 \to 3$ F D $3 * 2 \to 1$ C D $1 / 3 \to 2$ C G  2 G  | steps $1$ J $* 2 \to 2$ I $2$ I $* 3 \to 1$ F $1$ F $* 3 \to 3$ D $3$ D $* 2 \to 1$ C $1$ C $/ 3 \to 2$ G  2 G  |

Table 4: Example of a unit conversion task problem represented in different formats.

**Implementation.** To maintain consistent difficulty across our analyses, we use graphs with 10 nodes and 12 edges, and problems that could be solved using exactly 5 edge traversals. All arithmetic operations in this task are performed in modulo-5 to avoid the arbitrary fractions and large numbers that would result from compounding multiplication and division operations involved in multi-step problems. This allows us to focus on the reasoning component of the task rather than the numerical accuracy of performing long arithmetic operations. We use 4-layer transformers encoders for all our experiments in this task, which are trained using teacher-forcing on datasets of 10,000 randomly generated problems. We measure correctness by extracting the tokens between  and , which in fully trained models always consists of 1, 2, 3, or 4 followed by the goal unit, resulting in a 25% chance to correctly guess the answer, even with incorrect intermediary steps. More specific model details and comparisons can be found in Appendix Section B.1.

**Results.** All models successfully learned to generate sequences with the corresponding template, but the accuracy of the generated sequences varied from chance to nearly perfect across conditions. Our findings (Table 5) demonstrate foremost the importance of having the relational components as part of the target sequence, indicated by the near-chance accuracy of the numeric-only model when trained without planning, and the much higher success rate of all variants including abstract variables (variables corresponding to units).

Of the variants in which the model generates both relational and numeric output at test, the interleaved units-then-numbers model (RNRN) has the highest accuracy. Producing the relational plan first followed by numeric sequences (RRNN) is slightly worse, comparable to our findings in GSM8K-R. The fact that units-then-numbers is the best of the interleaved formats when the model does not first generate a relational plan suggests that identifying all of the relevant units that need to go in a numeric computation prior to performing that computation can be very helpful.

| Method | Accuracy |
|---|---|
| **Baseline** | |
| Answer only | 24.7 (1.1) |
| **Numeric Only** | |
| Numeric Sequences (NN) | 25.9 (1.1) |
| **Relational and Numeric** | |
| Relational plan then numeric (RRNN) | 71.1 (2.1) |
| Interleaved: units then numbers (RNRN) | **85.8** (1.1) |
| Interleaved: numbers then units | 69.3 (2.9) |
| Interleaved: integrated | 54.1 (3.0) |
| Plan + Interleaved: units then numbers | 72.5 (2.2) |
| Plan + Interleaved: numbers then units | 74.4 (1.7) |
| Plan + Interleaved: integrated | 77.1 (1.9) |
| **Relational (Prompted) and Numeric** | |
| Relational plan then numeric (RRNN) | 96.67 (2.5) |
| Interleaved: units then numbers (RNRN) | 99.9 (0.1) |
| Plan + Interleaved: units then numbers | 96.7 (1.2) |
| Plan + Interleaved: numbers then units | 95.5 (2.2) |
| Plan + Interleaved: integrated | 97.6 (1.1) |

Table 5: Unit conversion accuracy over 20 runs. Standard errors in parentheses.

Although training the model to produce both a relational plan and relational steps interleaved with numbers is helpful in numbers-then-units and integrated conditions, the reverse is true in the units-then-numbers condition, where asking the model to produce an initial relational plan actually reduces accuracy from 83% to 72%. This pattern of results suggests that generating the correct initial relational plan can itself be a challenge, and that an incorrect initial plan then interferes with performing the correct computations. Consistent with this interpretation, we find that all models trained to produce a relational plan do significantly better when given the ground truth plan as part of the prompt, reaching over 95% accuracy in all but the numeric-only models. Conversely, when the model uses an incorrect plan, its accuracy drops to near 20%. This suggests that the primary challenge of this task is not performing the correct arithmetic operations, but knowing which steps to take next. For a more detailed breakdown, see Appendix Section B.2.

| # Graph Nodes | # Graph Edges | # Solution Steps | Interleaved (RNRN) | Numeric Only (NN) | MT Numeric (RR\|NN) | MT Plan (RR\|NN) |
|---|---|---|---|---|---|---|
| 5 | 5 | 2 | 100.0 | 94.2 | 100.0 | 100.0 |
| 6 | 6 | 2 | 100.0 | 50.0 | 98.4 | 89.6 |
| 7 | 8 | 3 | 99.0 | 27.8 | 85.8 | 50.2 |
| 10 | 12 | 5 | 83.5 | 25.9 | 71.6 | 29.8 |

Table 6: Unit conversion results by difficulty. **MT Plan** indicates the percent of relational plans correctly traversed from the start to goal units by the multitask model. **MT Numeric** indicates final answer accuracy in the numeric only outputs by the multitask model.

**Limitations of numeric-only and multitask representations.** The near-chance performances of numeric-only (NN) and multitask (RR|NN) models are at odds with our results in GSM8K-R, as well as some other previous works that solved word problems by mapping them to arithmetic expressions first (Wang et al., 2017; Amini et al., 2019). Other than the synthetic nature of the UC task, one key distinguishing feature from GSM8K and other naturalistic math datasets is the relatively higher problem complexity. Consider the GSM8K problem shown in Table 2, which requires only a 2-step solution using just 6 unique quantity-unit pairs, and where the quantities invoked in the solution steps appear in the same order as presented in the prompt. In contrast, the graphs used in our analyses contain 10 nodes with 12 edges, and the relations are always presented in random order with no correspondence to how they appear in the solution. These features could make the unit conversion task more difficult, requiring more relational planning.

We test this hypothesis by training the numeric-only (NN), multitask (RR|NN), and interleaved units-then-numbers (RNRN) models on three easier datasets that contain problems involving smaller graphs with 5, 6, 7 nodes and only 2 to 3 solution steps. We find that while the RNRN models reach near perfect accuracy in all three problem complexities, the NN models only solve 94.2%, 50%, and 28% of the 5, 6, and 7 node problems respectively. Likewise, the RR|NN solves 100%, 89.6%, and 50.2% of the problems respectively, even though, interestingly, it produces correct plans 100%, 98.4%, and 85.8% of the time, indicating a weak transfer effect from learning to produce the plans to correctly solving the problems. In sum, while the numeric-only and multitask approaches may be effective on simpler problems, this strategy also does not scale well with problem complexity.

## 5 DISCUSSION

We find that relational reasoning is a key component of mathematical reasoning, whether using natural language or abstract symbols as indicated by our experiments on the GSM8K-R and the unit conversion tasks. Models trained with relational abstractions outperform models trained with numerical expressions only, and making these abstractions more salient improves performance further still. While the models can solve some problems without relational abstractions at test time, and can benefit from learning to generate the relational plan separately as in the multitask setup, performing both relational and numerical reasoning together scales far better with model complexity.

We also find that even when all the relational and numerical components are present, how they are ordered makes a significant difference. Among the variants we considered, performing the relational reasoning step just before the numerical computation step is most advantageous, outperforming cases where the full relational plan must be generated at the outset. Lastly, we find that providing the model with the correct abstract steps produces a massive boost in performance, resulting in a 3-fold increase in accuracy for the GSM8K-R task and near-ceiling accuracy in unit conversion, suggesting that the core of the challenge is indeed correct relational planning.

These results suggest that the popular approach to modeling mathematical reasoning through natural language datasets may be limited, and echo the conclusion in Hendrycks et al. (2021) that making significant strides in this domain will require a paradigmatic shift in how we understand the problem space. The diversity of problems in GSM8K-R and the consistency of results across both and the UC tasks provide confidence that relational abstractions are indeed central to mathematical reasoning. This points to an exciting future direction in understanding how relational abstractions can not only be used, but also identified by neural models, opening opportunities to engage with other math datasets such as MathQA (Amini et al., 2019) and MATH (Hendrycks et al., 2021) without the need for human annotations.We hope that our findings will motivate future research on the role of relational abstraction in mathematical reasoning, leading to deeper insight and stronger performance in this challenging and exciting domain.

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

| Method | Greedy | Simple Plurality | Verifier Rerank | Verifier Weighted Plurality |
|---|---|---|---|---|
| Original Solution | 17.44 | 21.53 | 19.86 | 23.35 |
| Our Equation Only | 22.97 | 23.58 | 23.96 | 24.79 |
| Socratic + Eqn. (aux-first) | 19.03 | 21.46 | 22.29 | 23.35 |
| Socratic + Eqn. (interleaved) | 21.00 | 22.21 | 25.85 | 25.47 |
| Relation + Eqn. (aux-first) | 19.48 | 22.97 | 22.75 | 25.55 |
| Relation + Eqn. (interleaved) | 22.97 | 26.31 | 25.63 | 29.49 |
| Relation + Eqn. (multitask) | **28.05** | **29.42** | **28.28** | **30.17** |

Table 7: GSM-8K Top-1 Test Accuracy (%) Using 20 Samples. **Bold** = Best Answer Format; Underline = Best Voting Mechanism. We take results from the best voting mechanism for each method in the main paper.

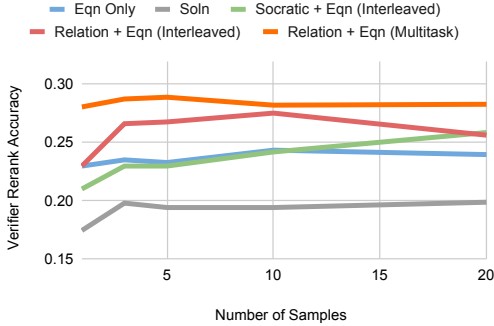

Figure 2: Verifier reranking accuracy

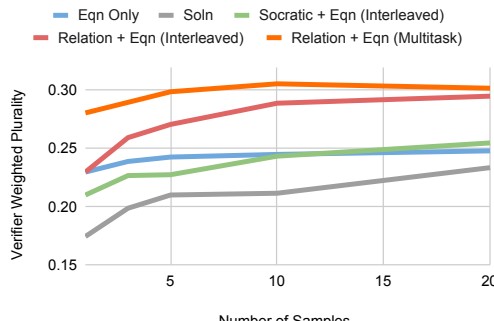

Figure 3: Verifier weighted plurality accuracy

# A  ADDITIONAL RESULTS

## A.1  GSM-8K RESULTS USING SAMPLES

In Table 7, we study more sample-based mechanisms for generating solutions. We generate 20 samples using softmax sampling (temperature = 0.9), and to aggregate the answers, we considered plurality voting (Wang et al., 2022) and the following verification-based techniques:

- **Verification.** As originally proposed in Cobbe et al. (2021), we train a separate verifier model using samples generated by our main model. The verifier takes as input the concatenated sequence of question and answer, then outputs a sequence of scores predicting whether the answer is correct or not. We generate the training samples using the main model after two epochs of fine-tuning, then fine-tune the GPT2-M model as our verifier.

- **Verifier weighted plurality.** We find that as the number of samples grows, a simple reranking mechanism performs worse as it has more incorrect options to choose from as the top choice. Cobbe et al. (2021) proposes using the voting mechanism to select the top-$K$ ranked samples as seeds and voting among these candidates. However, this requires a larger number of samples for the voting process, and moreover, $K$ becomes yet another hyperparameter to tune. Here, we explore a simpler approach of using the verifier score to weigh the votes. We find that it smooths out predictions and achieves higher accuracy.

All models seem to improve with using 20 samples, and our verifier weighted plurality is the best approach, achieve the best overall accuracy on all but one condition. Figure 2 and 3 show accuracy as a function of number of samples, and the verifier weighted plurality achieves higher scores with more samples.

Table 7 also indicates that performance of verification-based approaches benefits more from additional auxiliary information (whether in the form of natural language or abstract relations). For instance, our proposed *relation + equation (interleaved)* format has a similar performance to *equation only* using greedy decoding, but achieves significantly better performance with a verification voting procedure, while *equation only* receives a smaller boost (interleaved improved by +6.52% vs. equation only +1.82%). The *original solution* also receives a boost of +5.91%, except that the absolute accuracy is 6.14% lower than *relation + equation (interleaved)*, a rather wide gap. This dependence on a verification plus voting procedure suggests that relational abstraction is a more computationally demanding task that requires repeated processing of information.

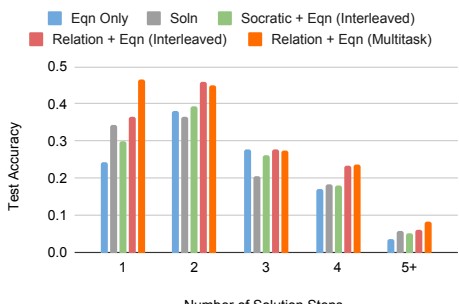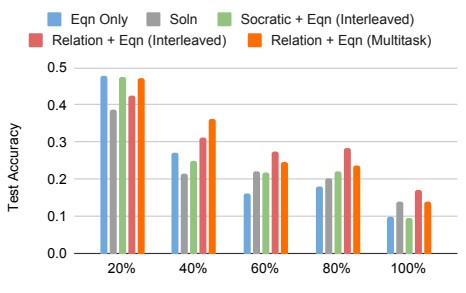

Figure 4: Accuracy vs. number of reasoning steps in the groundtruth answer.

Figure 5: Accuracy vs. percentile of solution length (percentiled separately by condition).

## A.2 GSM-8K RESULTS ON DIFFERENT SOLUTION LENGTH

In Figure 2 and Figure 3 we show the accuracy as a function of number of samples in both reranking and weighted plurality voting schemes. Reranking sometimes suffers from lower accuracy with more number of samples, whereas weighted voting has an overall positive trend as the number of samples go up.

We compare the performance of problems with different numbers of solution steps (Figure 4) and different generated sequence lengths (Figure 5). The overall trend confirms that models perform worse with longer answers. Figure 5 suggests that Equation Only tends to suffer from more degradation as the relative solution length increases.

Table 8: Hyperparameters of models in Table 9. All analyses reported elsewhere in the paper use Medium (M) hyperparameters.

| Size | # Parameters | Memory | # Layers | Layer Size | # Heads | Dim. Feedforward |
|---|---|---|---|---|---|---|
| Small (S) | 1.60M | 6.39 MB | 3 | 256 | 4 | 512 |
| Medium (M) | 2.12M | 8.49 MB | 4 | 256 | 4 | 512 |
| Large (L) | 3.18M | 12.714 MB | 6 | 256 | 4 | 512 |
| X-Large (XL) | 12.65M | 50.594 MB | 6 | 512 | 8 | 1024 |

Table 9: Comparison of final answer accuracy for each solution type with models of different sizes. Medium (M) contains averages of 20 models. Small (S), Large (L), and X-Large (XL) contain averages of 3 models.

| Size | Plan | Numeric Only | Interleaved: Units-Then-Numbers | Interleaved: Numbers-Then-Units | Interleaved: Integrated |
|---|---|---|---|---|---|
| S | Yes | 72% | 66% | 70% | 81% |
| S | No | 22% | 58% | 76% | 57% |
| M | Yes | 69% | 73% | 74% | 77% |
| M | No | 26% | 83% | 69% | 54% |
| L | Yes | 75% | 70% | 72% | 78% |
| L | No | 23% | 94% | 81% | 63% |
| XL | Yes | 66% | 62% | 82% | 67% |
| XL | No | 29% | 89% | 81% | 55% |

# B  UNIT CONVERSION

## B.1  MODEL DETAILS

All models used in the unit conversion experiments consisted of a linear token embedding layer, a transformer encoder, and a linear token decoder. We trained the models using teacher-forcing on datasets of 10,000 randomly generated problems with 20,000 gradient updates on batches of 256 samples. All experiments in the main manuscript were conducted using Medium (M) size models as detailed in Table 8.

We intentionally kept the model sizes small in the unit conversion tasks compared to the large language models used in the GSM8K dataset. Within the range of modest model sizes we tested, we observed the expected trend of increasing performance with larger models and consistent benefits from learning with relational abstractions. Table 8 lists the model hyperparameters and Table **??** lists the accuracy results for each model size for each solution format. We trained 3 separate models for each solution format for sizes S, L, and XL and 20 models for size M.

## B.2  RELATIONAL PLANNING AND ARITHMETIC ACCURACIES

To understand the sources of error in our models, we break down our metrics to whether the model correctly generated a valid plan and whether the plan is then correctly used in the numerical computations. Here, we define a valid plan as a series of steps involving just the units that all exist in the graph defined by the prompt and successfully connects the starting unit to the target unit. Tables 10 and 11 detail the accuracy results with rows representing the different solution formats and columns representing our different metrics of accuracy. Each cell reports the average accuracy using 20 separate models.

We describe the metrics as they appear in Tables 10 and 11.

1. Overall accuracy: given just the prompt, we check whether the model's final answer is correct

2. Accuracy using ground-truth plan: given the prompt and a correct plan, we check whether the final answer is correct

3. Plan accuracy: given just the prompt, we check whether the units correctly lead to the target unit, regardless of the numerical accuracy

4. Accuracy when model generated plan is correct: we check whether the model's final answer is correct on problems that the model generated a correct plan, and the model uses its own correct plan

5. Accuracy when model generated plan is incorrect: we check whether the model's final answer is correct on problems that the model generated an incorrect plan, and the model uses its own incorrect plan

6. Accuracy using ground-truth plan when model generated plan is incorrect: we check whether the model's final answer is correct on problems that the model generated an incorrect plan, but the model uses a given correct plan

## B.3 MODULUS

The use of a modulo space is useful for our UC experiments, but it is possible that it could produce unintended side effects. For example, using modulo-5 forces multiple conversion rules to use the same numbers. To test for this, we generate additional 10-node graph problems using modulo-23 and modulo-53 which would have lower chances of multiple rules using the same numbers in a given problem. We train 5 interleaved units-then-numbers (RNRN) and 5 numeric only (NN) models on these datasets. Raising the modulus to 23 and 53 increases difficulty, reducing the accuracy of the RNRN model to 71.0% and 31.6% respectively, but numeric-only accuracy drops further to 4.5% and 1.9%, i.e. the expected accuracies for randomly guessing.

Table 10: Unit conversion accuracy on training set.

| | Overall Accuracy | Accuracy Using Ground-Truth Plan | Plan Accuracy | Accuracy When Model Generated Plan is Correct | Accuracy When Model Generated Plan is Incorrect | Accuracy Using Ground-Truth Plan When Model Generated Plan is Incorrect |
|---|---|---|---|---|---|---|
| Interleaved | 71.14% | 96.67% | 66.15% | 96.76% | 21.04% | 96.51% |
| Plan + Numeric | 76.26% | 100% | 69.61% | 100% | 22.18% | 100% |
| Plan + Interleaved: Units Then Numbers | 74.03% | 100% | 66.96% | 100% | 21.27% | 100% |
| Plan + Interleaved: Numbers Then Units | 78.86% | 100% | 73.37% | 100% | 29.90% | 100% |
| Plan + Interleaved: Integrated | 85.84% | 99.91% | 82.30% | 99.93% | 20.30% | 99.84% |

Table 11: Unit conversion accuracy on test set.

| | Overall Accuracy | Accuracy Using Ground-Truth Plan | Plan Accuracy | Accuracy When Model Generated Plan is Correct | Accuracy When Model Generated Plan is Incorrect | Accuracy Using Ground-Truth Plan When Model Generated Plan is Incorrect |
|---|---|---|---|---|---|---|
| Interleaved | 71.14% | 96.67% | 66.15% | 96.76% | 21.04% | 96.51% |
| Plan + Numeric | 76.26% | 100% | 69.61% | 100% | 22.18% | 100% |
| Plan + Interleaved: Units Then Numbers | 74.03% | 100% | 66.96% | 100% | 21.27% | 100% |
| Plan + Interleaved: Numbers Then Units | 78.86% | 100% | 73.37% | 100% | 29.90% | 100% |
| Plan + Interleaved: Integrated | 85.84% | 99.91% | 82.30% | 99.93% | 20.30% | 99.84% |

## C  HUMAN ANNOTATOR INSTRUCTIONS

We include our instruction for human annotators for collecting the abstract relational plan data for GSM-8K dataset. The following pages contain an instruction as well as an example to be annotated with empty fillable boxes. This shows the user interface that the human annotators used when the labeling task was performed.

# Instruction

You will be assigned with some grade school math questions. The full solution is provided below each question. For most steps in the solution, there is a math equation being highlighted. Please add a line of *explanatory text* for each equation. The explanation should follow the *same format* as the original equation, while describing the items with short phrases that connect the equation with the relevant quantities mentioned in the problem and with quantities computed in other problems. Try to construct phrases that characterize the quantities succinctly while avoiding ambiguity and use the same phrase to refer to the same quantity a second time.

Here are some example questions. The purple text below illustrates the kinds of phrases that we ask you to fill in:

## Example #1

**Question:** Janet's ducks lay 16 eggs per day. She eats three for breakfast every morning and bakes muffins for her friends every day with four. She sells the remainder at the farmers' market daily for $2 per fresh duck egg. How much in dollars does she make every day at the farmers' market?

**Solution:** Janet sells 16 - 3 - 4 = <<16-3-4=9>>9 duck eggs a day.
She makes 9 * 2 = $<<9*2=18>>18 every day at the farmer's market.

Final answer: 18.

**Line 1:** 16-3-4=9          **Explanation:** Eggs laid - eggs eaten - eggs baked = eggs sold
**Line 2:** 9*2=18          **Explanation:** Eggs sold * price per egg = amount earned

*Note that we have preferred the use of very general names for variables such as "price" rather than "dollars" to encourage the recognition of common structures of variables.  We also used the exact phase 'eggs sold' both for the result of line one and for the same quantity when it occurred on the left hand side in line 2.*

*Please try to explain all quantities in the equations, including the item after the "=" sign.*

*Please also have white space before and after mathematical symbols like "+", "-", "\*", "/",  "=", etc.*

## Example #2

**Question:** Jen is planning to sell her root crops. She has 6 yams which can be sold at $1.5 each, 10 sweet potatoes that cost $2 each, and 4 carrots which cost $1.25 each. If she sells everything, how much will she earn?

**Solution:** Jen can earn $1.5 x 6 = $<<1.5*6=9>>9 for the yams.
She can earn $2 x 10 = $<<2*10=20>>20 for the sweet potatoes.
And she can earn $1.25 x 4 = $<<1.25*4=5>>5 for the carrots.
Therefore, she will earn $9 + $20 + $5 = $<<9+20+5=34>>34 if she sells everything.
Final answer: 34.

**Line 1:** 1.5*6=9          **Explanation:** Price per yam * number of yams sold = amount earned on yams
**Line 2:** 2*10=20          **Explanation:** Price per sweet potato * number of sweet potatoes sold = amount earned on sweet potatoes
**Line 3:** 1.25*4=5          **Explanation:** Price per carrot * number of carrots sold = amount earned on carrots
**Line 4:** 9+20+5=34          **Explanation:** Amount earned on yams + amount earned on sweet potatoes + amount earned on carrots = total amount earned

*Note that we don't repetitively mention the person's name (Jen) since it does not help resolve any ambiguity by mentioning her name.*

## Example #3

**Question:** John had a son James when he was 19. James is now twice as old as his sister Dora, who will turn 12 in 3 years. How old will John's youngest son, who was born when John was 32, be in 3 years?

**Solution:** Dora is 12-3=<<12-3=9>>9.
So James is 9*2=<<9*2=18>>18 years old
That means John is 18+19=<<18+19=37>>37
Johns youngest son is 37-32=<<37-32=5>>5 years old
So he will be 5+3=<<5+3=8>>8 in 3 years
Final answer: 8.

**Line 1:** 12-3=9      **Explanation 1:** Dora's age in three years - three years = Dora's age now
**Line 2:** 9*2=18      **Explanation 2:** Dora's age * ratio of James' age to Dora's age = James' age
**Line 3:** 18+19=37 **Explanation 3:** James' age + John's age when James was born = John's age
**Line 4:** 37-32=5      **Explanation 4:** John's age - John's age when his youngest son was born = John's youngest son's age
**Line 5:** 5+3=8      **Explanation 5:** John's youngest son's age + three years = John's youngest son's age in three years

*In this example, it is necessary to mention people's names to avoid ambiguity. Only do this when necessary.*

## Example #4

**Question:** Every hour Joanne has to collect the coins out of the fountain inside the mall. During the first hour, she collected 15 coins. For the next two hours, she collected 35 coins from the fountain. In the fourth hour, she collected 50 coins from the fountain but she gave 15 of them to her coworker so she could buy a soda. How many coins did she have after the fourth hour?

**Solution:** 15 coins collected in hour one
35 coins collected in hour two
35 coins collected in hour three
50 coins collected in hour four
Before giving her coworker some coins there were 15+35+35+50=<<15+35+35+50=135>>135 coins
The number of coins after given 15 to her coworker is 135-15=<<135-15=120>>120
Final answer: 120.

**Line 1:** 15 coins collected in hour one      **Explanation:** Coins collected in hour one
**Line 2:** 35 coins collected in hour two      **Explanation:** Coins collected in hour two
**Line 3:** 35 coins collected in hour three      **Explanation:** Coins collected in hour three
**Line 4:** 50 coins collected in hour four      **Explanation:** Coins collected in hour four

**Line 5:** 15+35+35+50=135          **Explanation:** Coins collected in hour one + coins collected in hour two + coins collected in hour three + coins collected in hour four = total coins collected.

**Line 6:** 135-15=120          **Explanation:** Total coins collected - coins given to the coworker = coins remaining

*Note that not all lines will contain an equation, and in this case try to explain each solution line with plain words.*

*In some cases, as with the first four lines here, the explanation may repeat the content of the Line, but we ask you to provide such explanations, as in the example.*

## Equations with unknown variables

For each problem, before you can enter explanations, there will be a **required question** asking whether any of the lines of the solution contain unknown variables. In the example below, "C" is the unknown variable. If there are unknown variables, then please answer "yes" for the first question, and follow the example below to provide an explanation for each line.

**Question:** Farmer Brown has 20 animals on his farm, all either chickens or cows. They have a total of 70 legs, all together. How many of the animals are chickens?

**Solution:** Let C be the number of chickens.
There are 20-C cows.
The cows have 4*(20-C) legs.
The chickens have 2C legs.
The total number of legs is 2C+4(20-C)=70.
2C+80-4C=70
2C=10
C=<<5=5>>5.
Final answer: 5.

In this case, you will be asked to provide explanations for each line of the solution which will be displayed.  These lines will not simply be an equation as in other cases.  As before, the purple text shows the kind of explanation we are asking you to provide.

**Line 1:** Let C be the number of chickens.          **Explanation 1:** Define a variable for the number of chickens

**Line 2:** There are 20-C cows.          **Explanation 2:** Number of animals - number of chickens =

number of cows

**Line 3:** The cows have 4*(20-C) legs.  **Explanation 3:** Number of legs each cow * number of cows = number of cow legs

**Line 4:** The chickens have 2C legs.  **Explanation 4:** Number of chicken * legs per chicken = number of chicken legs

**Line 5:** The total number of legs is 2C+4(20-C)=70.  **Explanation 5:** Number of chicken legs + number of cow legs = total number of legs

**Line 6:** 2C+80-4C=70.  **Explanation 6:** Simplify toward finding the number of chickens

**Line 7:** 2C=10.  **Explanation 7:** Combine like terms toward finding the number of chickens

**Line 8:** 5=5  **Explanation 8:** Divide by 2 to determine the number of chickens

*Note that we have asked you to restate the quantity referenced by the variable and also to use the quantity, not the variable itself in your explanations.*

Collapse Instructions

**Question:** Cynthia has four times as many water balloons as her husband, Randy. Randy has only half as many water balloons as his daughter, Janice. If Janice throws all 6 of her water balloons at her father, how many water balloons does Cynthia have, which she could also choose to throw at Randy?

**Solution:** Randy has only half as many water balloons as Janice's 6, for a total of (½)*6=3 water balloons.
Cynthia has 4 times as many water balloons as Randy, for a total of 4*3=<<4*3=12>>12 water balloons
Final answer: 12

Does the solution seem correct?
○ Yes
○ No

Does the solution contain equations of unknown variables? (See instruction for an example of equations of unknown variables)
○ Yes
○ No

(If the line is empty, please skip the response)

**Line 1:** Randy has only half as many water balloons as Janice's 6, for a total of (½)*6=3 water balloons.

**Explanation 1:**

```

```

(If the line is empty, please skip the response)

**Line 2:** 4*3=12

**Explanation 2:**

```

```

(If the line is empty, please skip the response)

**Line 3:**

**Explanation 3:**

```

```

(If the line is empty, please skip the response)

**Line 4:**

**Explanation 4:**

```

```

(If the line is empty, please skip the response)

**Line 5:**

**Explanation 5:**

```

```

(If the line is empty, please skip the response)

**Line 6:**

**Explanation 6:**

```

```

(If the line is empty, please skip the response)

**Line 7:**

**Explanation 7:**

(If the line is empty, please skip the response)

**Line 8:**

**Explanation 8:**

(If the line is empty, please skip the response)

**Line 9:**

**Explanation 9:**

(If the line is empty, please skip the response)

**Line 10:**

**Explanation 10:**

(If the line is empty, please skip the response)

**Line 11:**

**Explanation 11:**

(If the line is empty, please skip the response)

**Line 12:**

**Explanation 12:**

**Additional comments**

