# OpenReview forum: "Learning to reason with relational abstractions"
_ICLR.cc/2023/Conference — Submitted to ICLR 2023_

### Official Review · Reviewer_XGS4 · 2022-10-19

**Confidence:** 2
**Correctness:** 3
**Technical Novelty And Significance:** 2
**Empirical Novelty And Significance:** 2
**Recommendation:** 3

**Clarity, Quality, Novelty And Reproducibility:**

The method section of this paper is not clear. The results are not significant.

**Strength And Weaknesses:**

Strength:

1. This paper proposes to decompose the problem solving process into relational abstraction and arithmetic expression, which is amenable to large language models and is a middle-ground between neurosymbolic, natural language-based, and arithmetic-only approaches.

2. This paper finds relational abstraction improves problem solving performance when supplied either during training or testing, making it a crucial component to study separately.

Weaknesses:

1. The key contribution, i.e., relational abstraction, are not described clear. The author should provide a general methodology that are formal and easy to follow.

2. The experiments are not comprehensive. This paper only uses one dataset for validating the ideas, which is far from enough.



**Summary Of The Paper:**

This paper finds that relational reasoning is a key component of mathematical reasoning, whether using natural language or abstract symbols as indicated by our experiments on the GSM8K and the unit conversion tasks. Training the models with relational abstraction can outperform models trained using numerical expressions only, and making these abstractions more salient improves performance
further still. It also find that even when all the relational and numerical components are present, how they are ordered makes a significant difference. Among all the variants, performing the relational reasoning step just before the numerical computation step is most advantageous, outperforming cases where the full relational plan must be generated at the outset.

**Summary Of The Review:**

More efforts required on the methodology and experiment sections.

---

> ### Author Response · Authors · 2022-11-19
> **Thank you for your comments**
>
> Thank you for reviewing our paper and your feedback. We have written a general response to address broader concerns as a separate comment, and we request that you read that before proceeding to read the remainder of our response regarding specific details and analyses raised in your review.
>
> >**The key contribution, i.e., relational abstraction, are not described clear. The author should provide a general methodology that are formal and easy to follow.**
>
> Thank you for the comment. We have made a major effort rewriting and reorganizing the sections. We have now dedicated Section 2 to describe our relational abstraction framework. Relational abstraction is a plan outlining the reasoning process without invoking any numbers. In the example of Janet selling eggs, “number of eggs laid”, “number of eggs eaten”, “number of eggs used in baking”, “remaining eggs” and “price per egg” are quantities needed to reach the target quantity, and the abstract relational plan specifies the steps that must be applied to the given quantities to reach the relevant intermediate quantities, and then applied to these quantities to reach the final answer.
>
> >**The experiments are not comprehensive. This paper only uses one dataset for validating the ideas, which is far from enough.**
>
> Our contribution is to present a new way of approaching reasoning problems. Prior approaches rely on annotations that do not have relational abstraction. In order to demonstrate the benefit, we collected annotations in our augmented GSM8K-R dataset and designed a synthetic unit conversion task. We believe they cover a good range of problems in real applications. Since no other datasets have studied our relational abstraction format, it is not possible to add another dataset.
>
>
> >**The method section of this paper is not clear. The results are not significant.**
>
> We apologize for any parts that were unclear, and we have made major revisions throughout the paper to clarify our methodology and bring out the key findings of our experiments. We hope that you would find our methods easier to understand and incorporate the newer version into your evaluation. Our new combined Table 1 at the beginning of the Experiments section indicates that after fine tuning, prompting with our new relational abstraction sequences can lead to dramatic gains on GSM8K.  Even more strikingly, in our UC task, our model’s performance goes from near chance when trained to generate numerical sequences only to levels as high as 83% when they generate the relational sequences themselves.  By analyzing the relational plan outputs of the models we have found that the models are near ceiling levels on those trials where they generate the plan sequence correctly.

---

### Official Review · Reviewer_wEws · 2022-10-21

**Confidence:** 4
**Correctness:** 4
**Technical Novelty And Significance:** 3
**Empirical Novelty And Significance:** 3
**Recommendation:** 6

**Clarity, Quality, Novelty And Reproducibility:**

Find below some clarification that are required to make this a better paper:

**clarity (c1)** : “relational abstraction” is mentioned 9 times in the introduction, yet it is not clear what is meant by it. An illustrative example at the beginning of the paper would probably clarify what the authors mean by “relational abstraction”. Another possibility is to rephrase the sentences and try to explain what is meant by saying something like “*giving logical formulas as intermediate step before the final answer*”

**clarity (c2)** : The paper introduces 4 ways of creating intermediate steps before an answer in Figure 1: A, B, C, D. The paper would be a lot more clear if the same references were used throughout the paper. For instance, in Table 2 and Table 4 it is not clear which row refers to which of these A, B, C, D techniques. Adding these labels in each row could potentially make the results easier to read.


**Strength And Weaknesses:**

**strengths** : This work explores different ways of combining mathematical reasoning into large language models, which offers insight into what can improve current models performance.

**weakness (w1)** : However, the paper is not easy to follow as some concepts are not explicitly defined, and the methodology and experimental results are difficult to parse (see the clarity section below)

**weakness (w2)** : Results so far only cover the final answer accuracy. The paper should make an effort to also measure the accuracy of the intermediate reasoning steps being generated during test time. This would tell if the model is actually able to generate interpretable reasoning steps that are useful to answer the final question.

**question** : In the unit conversion task, how does the numbers in Table 4 compare with the baseline model that tries to output the final answer directly? Table 4 should have an extra line for the baseline model.

**suggestion** : It would be nice to show that the conclusion found in this work holds true for other mathematical datasets and not just for GSM8k. Other datasets could include: AQuA-Rat (Ling et al., 2017), MATH Dataset (Hendrycks et al., 2021), MathQ (Drori et al., 2021).
Similarly, previous work (Gontier et al., 2020) has also explored what type of intermediate steps are better to train language models to perform multi-step reasoning. Discussing how this work differs could benefit the paper.


**Summary Of The Paper:**

This paper defines 4 different ways to fine-tune language models with intermediate reasoning steps for multi-step mathematical reasoning datasets. After being asked a question requiring multiple symbol manipulations, a model can:
 - Baseline: output the answer directly
 - A : output the chain of numerical expressions before the final answer
 - B : output the chain of logical expressions followed by the chain of numerical expressions before the final answer
 - C : output the chain of logical expressions interleaved with the chain of numerical expressions before the final answer, or
 - D : output either the chain of logical expressions or the chain of numerical expressions before the final answer, based on some prompt at test time.

Experimental results on GSM8k and a synthetic task demonstrate that fine-tuning models with these intermediate steps (in particular methods B, C, D) improves the overall answer accuracy of the model.


**Summary Of The Review:**

Overall, this is an interesting paper that shows what type of logical formulation can be used to improve language models to solve multi-step mathematical problems.
However the paper could benefit from some clarifications (see **w1** referring to **c1** **c2** in the sections above) and some extra analysis (see **w2** in the sections above).

---

> ### Author Response · Authors · 2022-11-19
> **Thank you for your comments**
>
> Thank you for reviewing our paper and your feedback. We have written a general response to address broader concerns as a separate comment, and we request that you read that before proceeding to read the remainder of our response regarding specific details and analyses raised in your review.
> >**Weakness (w1): However, the paper is not easy to follow as some concepts are not explicitly defined, and the methodology and experimental results are difficult to parse (see the clarity section below)**
>
> Thank you for this feedback. We made significant revisions to our manuscript to clarify important concepts, our methodology, and the experimental results. We reorganized the tables to offer clearer side-by-side comparisons of our results across both tasks, which we hope will help communicate the motivations and insights from our experiments.
>
> >**Weakness (w2): Results so far only cover the final answer accuracy. The paper should make an effort to also measure the accuracy of the intermediate reasoning steps being generated during test time. This would tell if the model is actually able to generate interpretable reasoning steps that are useful to answer the final question.**
>
> We thank the reviewer for this suggestion.  We have added these results for the UC tasks to the manuscript, which can now be found in Tables 1, 10, and 11. Unfortunately, we can only calculate this metric for the UC task because there may be more than one way to indicate an item (e.g. ‘blue ties’ and ‘red ties’, or just ‘ties’? ‘2 dozen eggs’ or ‘24 eggs’?). In fact, this was one of the most significant challenges in working with human annotators for the GSM8K-R annotations.
>
> >**Question: In the unit conversion task, how does the numbers in Table 4 compare with the baseline model that tries to output the final answer directly? Table 4 should have an extra line for the baseline model.**
>
> Training the model to output the final answer directly yields an accuracy 24.7% (no different than chance = 25%). We have added this result to our table.

---

> > ### Author Response · Authors · 2022-11-19
> > **Part 2**
> >
> > >**Suggestion: It would be nice to show that the conclusion found in this work holds true for other mathematical datasets and not just for GSM8k. Other datasets could include: AQuA-Rat (Ling et al., 2017), MATH Dataset (Hendrycks et al., 2021), MathQ (Drori et al., 2021). Similarly, previous work (Gontier et al., 2020) has also explored what type of intermediate steps are better to train language models to perform multi-step reasoning. Discussing how this work differs could benefit the paper.**
> >
> > As much as we would like to see our insights applied to other datasets, we leave this important task to future research. We chose the GSM8K dataset because it offered a diverse set of problems that are relatively easy for humans to solve, but is still challenging for neural networks. Our goal in this paper was not to beat benchmarks for any given datasets, but to understand how we should think about formal and mathematical reasoning, and we believe that our results as presented in our paper using both natural and synthetic datasets demonstrate general principles that would be broadly applicable.
> >
> > We thank the reviewer for the suggestion to compare our work to that of Gontier et al.  In fact or UC task offers substantial advantages compared to their approach.  As we explain more briefly in the Related Work section, our UC approach parallels that of Gontier et al but with a crucial difference.  These authors investigated logical reasoning over a fixed data-base of specific relational facts, (e.g. Natasha is the grand-daughter of Betty, Florence is Gregorio's sister, Gregorio is a brother of Natasha), training models to produce an inferable relation to a probe question, and found only small advantages of a plan sequence compared to generating the answer directly. (Q: Who is Florence to Betty? A: Granddaughter).  In contrast, our UC task affords separating the abstract relational plan from the specific numerical computations.  We compare learning from just the specific numerical expressions needed to solve the problem with learning to generate the abstract relational plan separately from the numerical statements themselves, and we present a new data-base of quantitative relations that the model must use as premises to accompany each inference problem, so that the model cannot memorize a fixed set of specific transformation rules.  This allows us to demonstrate a striking advantage from learning to produce the abstract relational sequence rather than just the necessary numerical expressions. Gontier et al found high levels of accuracy in their answer-only baseline and observed a benefit for proof sequences only when the proof, which always contained the to-be-inferred relational statement, was provided as a prompt. In contrast, we use conditions where models are at chance when prompted to generate the answer from the problem statement itself and only slightly exceed chance when trained only to generate only numerical steps.  Furthermore, our relational plans do not contain the numerical information needed to obtain the correct answer.  In these conditions, we show dramatic benefits when the models have learned to generate relational plans for themselves: When prompted with the plan, or when they generate the plan correctly, these models jump from near chance to near-ceiling accuracy.
> >
> > >**Clarity (c1) : “relational abstraction” is mentioned 9 times in the introduction, yet it is not clear what is meant by it. An illustrative example at the beginning of the paper would probably clarify what the authors mean by “relational abstraction”. Another possibility is to rephrase the sentences and try to explain what is meant by saying something like “giving logical formulas as intermediate step before the final answer”.**
> >
> > Thank you for the comment and suggestion. We made a major effort to clarify our presentation of this idea. We have now dedicated Section 2 to describe our relational abstraction framework, and we have integrated your suggestion of logical formulas and intermediate steps.
> >
> > >**Clarity (c2) : The paper introduces 4 ways of creating intermediate steps before an answer in Figure 1: A, B, C, D. The paper would be a lot more clear if the same references were used throughout the paper. For instance, in Table 2 and Table 4 it is not clear which row refers to which of these A, B, C, D techniques. Adding these labels in each row could potentially make the results easier to read.**
> >
> > Thank you for this suggestion. We have modified our tables to present a more unified view of our results across both the GSM8K-R and UC tasks, and have also revised our terminology to be more consistent as well.  We hope these changes bring out the common themes in our results across the two data sets, and help clarify how the two studies both support and complement each other.

---

> > > ### Comment · Reviewer_wEws · 2022-11-21
> > > **Response to authors**
> > >
> > > Thanks for your clarifications and answers. The revised version of the paper is more clear now. Score updated to 6.
> > > - - -
> > >
> > > **w1**, **c1**, **c2**: thank you for addressing these clarifications. The paper is easier to follow now.
> > >
> > > **w2**: thank you for adding this information. However I notice that Table 10 (UC train) and 11 (UC test) have the exact same numbers in them... I suppose this is a copy-paste mistake? Please explain / update.
> > >
> > > **previous suggestion**:  _"separating the abstract relational plan from the specific numerical computations"_ is indeed an interesting differentiation. Thanks for making this clear.
> > > - - -
> > >
> > > **new suggestion**
> > >
> > > As the authors mentioned in response to **w2**, working with human annotations to label the abstract relational plan of GSM8k was challenging. The fact that humans can express the same item in different ways makes me wonder how "abstract" is really the human annotations. In formal logic all natural language expressions referring to the same item would have been labeled with the same unique variable. The fact that it is not the case in this work (due to human annotations) remains a small weakness. It should be acknowledged in the discussion of the paper (unless I've missed it in the revised version?).
> > > Nevertheless, as imperfect as human annotations can be, this work still shows some advantages to separating the abstract relational plan from the specific numerical plan.

---

> > > > ### Author Response · Authors · 2022-11-22
> > > > **Response to Reviewer 3**
> > > >
> > > > Thank you for this comment.  Indeed we completely agree that the human annotations retain some ambiguities that formal notation avoids, and indeed this may be one of the reasons why the performance when they are provided still has room for improvement.  We have now added a sentence (marked in << >>) at the end of the second paragraph of the discussion to acknowledge this point, reprinted below.
> > > >
> > > > We also find that even when all the relational and numerical components are present, how they are
> > > > ordered makes a significant difference. Among the variants we considered, performing the relational
> > > > reasoning step just before the numerical computation step is most advantageous, outperforming
> > > > cases where the full relational plan must be generated at the outset. Lastly, we find that providing
> > > > the model with the correct abstract steps produces a massive boost in performance, resulting in a
> > > > 3-fold increase in accuracy for the GSM8K-R task and near-ceiling accuracy in unit conversion,
> > > > suggesting that the core of the challenge is indeed correct relational planning. <<We note that our human
> > > > annotations may more ambiguous than the variables in an abstract formal system, and this may be
> > > > one factor contributing to the greater difficulty our models have in extracting and using these relations
> > > > to solve the GSM8K problems.>>
> > > >
> > > > As for the error in Tables 10 and 11, thank you for pointing out this mistake. The data shown for both tables was for the test set. The accuracy measures are near 100% for all cells except for "Accuracy When Model Generated Plan is Incorrect", which are 0%, and a few cells with "Generated Plan is Incorrect" that are NA because all plans were correct.
> > > >
> > > > We are not able to update the PDF at this point, and these changes will be included in the final manuscript upon publication.

---

### Official Review · Reviewer_YJ77 · 2022-10-25

**Confidence:** 3
**Correctness:** 3
**Technical Novelty And Significance:** 1
**Empirical Novelty And Significance:** 2
**Recommendation:** 3

**Clarity, Quality, Novelty And Reproducibility:**

The paper is overall clearly written and provides some details for reproducing the results.

**Strength And Weaknesses:**

One major concern about the proposed approach is that annotating the training examples with relational/numeric representations requires significant human efforts. Though the authors’ claim: “the challenge that model face lies primarily in constructing the relational plan” is indirectly supported by their experiments, they do not provide a very practical solution for tackling this problem. Regarding ablation studies, the authors do not provide enough insight on why certain ways of presenting the annotated relational and numeric sequences help the language models to perform better.

The authors used GPT2-M and GPT2-XL for solving GSM8K; however, for the synthetic setting, the authors instead use a 4-layer transformer encoder with no pre-training and the motivation for this model choice is unclear.

**Summary Of The Paper:**

This paper studies the problem of how to improve language models’ ability to perform reasoning. In particular, the authors focus on the task of mathematical reasoning: on both GSM8K and a synthetic task called unit conversion, the authors showed that by training models to generate abstract representations (referred to as relational and numerical in paper) of the reasoning steps, the models perform better at solving reasoning problems. The authors also conduct ablation studies on how the order of relational/numerical sequences affects model performance. The abstract representations for GSM8K are annotated by humans and that for the unit conversion task are automatically generated.

**Summary Of The Review:**

The authors show by experiments that incorporating abstract relational representations can help language models to perform reasoning, but they do present an effective approach to accomplish that. Apart from the major claim of the paper, the analysis/ablation studies do not provide any other insight into this problem.

---

> ### Author Response · Authors · 2022-11-19
> **Thank you for your comments**
>
> Thank you for reviewing our paper and your feedback. We have written a general response to address broader concerns as a separate comment, and we request that you read that before proceeding to read the remainder of our response regarding specific details and analyses raised in your review.
>
> >**One major concern about the proposed approach is that annotating the training examples with relational/numeric representations requires significant human efforts.**
>
> We acknowledge this as a legitimate limitation for anyone that wishes to implement our strategy as a practical solution towards improving math problem solvers. However, our goal in this paper is not to offer a practical solution but rather an investigation into how transformer-based models learn. We discuss this more in-depth in our general response.
>
> >**Though the authors’ claim: “the challenge that model face lies primarily in constructing the relational plan” is indirectly supported by their experiments, they do not provide a very practical solution for tackling this problem.**
>
> As we note in our broader response, we do not present this research as a practical    solution for math datasets, but as a means to diagnose and understand why they are challenging for neural networks, which is an important step towards finding possible solutions. This paves the way for exciting future directions towards building better models for mathematical reasoning.
>
> We substantiate this claim that the primary challenge in these mathematical reasoning tasks is in constructing the relational plan by comparing the model’s performance when it generates its own plan to when the ground-truth plan is provided to the model. If indeed the main challenge is in generating the plan, then we should expect that the model’s accuracy would be much higher when the ground-truth plan is provided. We find strong evidence that this is the case, observing up to a threefold increase in accuracy from 22.97% to 66.26% in the Relation + Eqn. (interleaved) model. Similar results are obtained in the UC task where the models consistently reach 100% accuracy in all but the Numeric-Only model, which still reaches 96.67.
>
> We have conducted a follow-up analysis using the UC dataset comparing the effect of ground-truth prompting between problems that the model generates a correct plan for and problems that it does not. We find that the accuracies are nearly identical between these two conditions, suggesting that once the model has a correct relational plan, regardless of whether or not it could generate it itself, it can then solve the problem with near perfect accuracy. We have revised our manuscript to emphasize this point in the main text, and we also include a complete table of results in the Appendix Tables 10 and 11.
>
> >**Regarding ablation studies, the authors do not provide enough insight on why certain ways of presenting the annotated relational and numeric sequences help the language models to perform better.**
>
> We admit that we do not fully understand why some relational formats do better than others, but we offer some ideas in our paper as to why these patterns emerge.
>
> First, we note from our ground-truth prompting experiments that the primary challenge for these models is indeed constructing the relational plan, and models generally succeed in following through with the arithmetic computation given a correct plan. This holds for both  ‘abstract then arithmetic’ and ‘interleaved - units then numbers’ formats in both GSM8K-R and UC datasets. This may be why ‘interleaved units then numbers’ outperforms the other two interleaved conditions since it is only in this format that the full abstract relation is stated before the arithmetic.
>
> Second, the ‘interleaved - integrated’ condition is unique in that the units are not directly adjacent to each other, which may hinder the model’s ability to learn them as pairs.
>
> Lastly, we find that in general, these transformer-based models struggle with longer sequence lengths, as evidenced by Figure 5 in our appendix, where test accuracy decreases as solution lengths increase. This may be why ‘Interleaved: units then numbers’ condition decreases in accuracy when the relational plan is prepended to the solution: it increases the solution length, creating more room for error, without adding any further useful information for the model.

---

> > ### Author Response · Authors · 2022-11-19
> > **Part 2**
> >
> > >**The authors used GPT2-M and GPT2-XL for solving GSM8K; however, for the synthetic setting, the authors instead use a 4-layer transformer encoder with no pre-training and the motivation for this model choice is unclear.**
> >
> > Our goal in the UC task was to strip the task to its minimum elements, removing the complexities of natural language and any influence from the massive datasets that large language models are trained with. We, therefore, intentionally trained our models from scratch so that we could verify that relational abstractions directly impact learning to reason and have confidence that there is no hidden interaction with the LLMs’ pretraining that could potentially confound our analyses. We also intentionally kept our model sizes modest to demonstrate that relational abstractions help learning in both large models as shown through our GSM8K-R experiments and in smaller models. We have clarified our motivations in the manuscript.
> >
> > The use of 4-layer transformers was somewhat arbitrary, based on preliminary experiments in which we found no important differences. We added to the Appendix Tables 8 and 9 which details performances for models with 3 layers (1.60M parameters) to 6 layers (12.65M parameters). Unsurprisingly, larger models do generally perform better than smaller ones, but this is tangential to our main interest, which is that all models we tested performed significantly better with relational abstractions than not.

---

### Official Review · Reviewer_cUHL · 2022-10-27

**Confidence:** 4
**Correctness:** 3
**Technical Novelty And Significance:** 3
**Empirical Novelty And Significance:** 3
**Recommendation:** 6

**Clarity, Quality, Novelty And Reproducibility:**

Clarity, Novelty And Reproducibility are good!
Quality of the work needs to be improved.

**Strength And Weaknesses:**

Strengths:
1. A very simple idea but brings a large performance improvement
2. Writing and presentation is very clear

Weaknesses:
1. The authors only tried very large models, I would like to see how this method scales w.r.t. model sizes.  Does it only work on larger models?
2. The results on the unit conversion dataset can benefit from more analyses.  For example, after adding the relational abstractions, there are not any natural language involved, and it is just a completely symbolic task.  Therefore, can the units be replaced with any number?  Or the units actually have meaning in the GPT models?  Would be interesting to figure out.
3. more datasets are needed.

**Summary Of The Paper:**

1) This paper proposes to incorporate relational abstractions into mathematical reasoning.  Hand-crafted a number of ways to combine relational abstractions and equations.
2) This paper presents empirical evaluations on GSM-8K and a unit conversion dataset.

**Summary Of The Review:**

Very well written paper, simple idea, strong experimental results

---

> ### Author Response · Authors · 2022-11-19
> **Thank you for your comments**
>
> Thank you for reviewing our paper and your feedback. We have written a general response to address broader concerns as a separate comment, and we request that you read that before proceeding to read the remainder of our response regarding specific details and analyses raised in your review.
>
> >**The authors only tried very large models, I would like to see how this method scales w.r.t. model sizes. Does it only work on larger models?**
>
> Our experiments for the GSM8K-R dataset were conducted using GPT2 variants, such as GPT2-XL, which is fairly modest in size (1.5B) by today’s standards for large language models (LLM) (e.g. GPT3 is 175B).
>
> The unit conversion (UC) experiments were conducted using fairly lightweight models, only 8.49MB in size, containing 2.12M parameters, and requiring a total of about 3 GB while training. We have also subsequently run a comparison of model sizes ranging from 1.60M parameters to 12.65M parameters, and found a consistent improvement in using abstract relations (see Appendix Table 9).
>
> Collectively, across the tiny models used in our UC experiments to the massive GPT2-XL model used in GSM8K, we find that abstract relations are important contributors regardless of model size.
>
> >**The results on the unit conversion dataset can benefit from more analyses. For example, after adding the relational abstractions, there are not any natural language involved, and it is just a completely symbolic task. Therefore, can the units be replaced with any number? Or the units actually have meaning in the GPT models? Would be interesting to figure out.**
>
> We share this concern regarding the GSM8K-R task, especially since it can only be reasonably solved using massive LLMs such as GPT2 and GPT3, and this was our motivation for the unit conversion experiments. We note that all models were trained from scratch in the UC task and using abstract tokens such as ‘A’, ‘B’, ‘C’, ‘D’, etc. guarantees that our units are completely devoid of semantic content.
>
> Moreover, because our UC datasets were generated using modulo-prime spaces, e.g. mod-5, even the arithmetic rules used in the task better resemble arbitrary mappings than the standard arithmetic that we use for real numbers.
>
> >**more datasets are needed.**
>
> While we agree that future work should extend our approach to other data sets, the  two datasets we do have, one naturalistic (GSM8K-R) and one synthetic (UC), should cover the broad range of bases for which this form of reasoning is applicable. We discuss this more in-depth in our general response.

---

### Author Response · Authors · 2022-11-19
**To all reviewers**

We thank all reviewers for their insightful and constructive comments. We have revised the paper extensively in response, and we have indicated most of our significant changes using blue text.

We begin by highlighting the key contribution of our work:  It is to show how powerfully multi-step mathematical reasoning can be facilitated by training a model to produce and then to use the abstract relationships underlying the numerical computations needed to solve the problem.  Multi-step mathematical reasoning is still a major challenge: In spite of recent improvements, even the strongest recent models solve less than 50% of the problems in many of the benchmark sets.   The two tasks in our paper serve as case studies that help demonstrate how using abstract relationships can help.

Our contribution is not a particular solution strategy, as though the GSM8K and the UC tasks were particular milestone problems to be solved. Nor is it an architectural modification to transformers or simple augmentation techniques that can be programmatically applied to any dataset. Instead, what we present here is an investigation into how algorithmic reasoning can be learned, and we view the diagnosis of current limitations and a clearer understanding of them equally important and meaningful in its own merit. Automatically extracting relational components from reasoning problems would be an exciting future direction given the analyses we present in this paper, and we note this solution-focused research as next steps.

A common comment was that our paper would be strengthened by the inclusion of more datasets.  Extending the approach to the more advanced problems found in other data sets such as those mentioned by the third reviewer would require an extensive materials development project and would certainly be a worthwhile next step.  That said, GSM8K, involving natural language and different types of problems, is already a highly diverse dataset, and we believe we have made a significant contribution by providing our new relational and numeric GSM8K-R data set.  This combined with the demonstrated dramatic advantages of relational abstractions offered by our findings from the UC task support our view that our findings are sufficiently rich and general to merit the attention of the community.
In the revision, we clarify what we mean when we speak of these abstract relationships, highlight the way the two experiments parallel and complement each other (using more consistent terminology across the two experiments as suggested be one of the reviewers and showing highlights of our findings in the new table 1), and add new analyses assessing accuracy of the relational planning in the UC model.  This brings out the striking way in which relying on relational abstractions can improve performance. We would like to emphasize that we do not intend the specific approaches we have used as practical ways of improving on the state of the art.  Instead, they serve to emphasize how important relying on relational abstractions can be.   If it comes to the attention of the community, we believe our work will inspire others to join us in exploring how to achieve this.

---

### Decision · Program_Chairs · 2023-01-20

**Decision:**

Reject

**Justification For Why Not Higher Score:**

The idea behind the paper is interesting, but presentation needs to be improved to convey it properly. Experiments do not fully support the claims.

**Justification For Why Not Lower Score:**

N/A

**Metareview: Summary, Strengths And Weaknesses:**

This work investigates how language models trained to perform mathematical reasoning can benefit from abstract representations during training. Specifically, the authors devise a set of heuristics to group intermediate, task-specific, relational and numeric templates for reasoning problems. To this end they propose a human-annotated variant of the GSM8K dataset that provides the ground truth labels to train the
model with these sequences of abstractions.

Reviewers had mixed opinions about the proposed contribution and highlighted how the proposed methodology is of limited scope as requires precise human-provided annotations for the task at hand to provide enough supervision. The fact that models trained on these augmented variant outperform baselines is comforting, as it is expected. I agree with the authors that the core contribution is not improving the state-of-the-art in mathematical reasoning, but shedding a light on what large language models can benefit from in terms of abstractions.
However, it is not clear how the results obtained with this variant of the usual mathematical reasoning benchmark and on the synthetic task might be generalizable to other tasks or real-world scenario easily. Lastly, presentation of the motivation, benchmark and experiment execution was leaving out important details and discussions. The authors did a good job improving it during the rebuttal. The two aforementioned concerns about applicability and generalizability of the claims, however, remain.

The paper is therefore rejected.



**Summary Of Ac-Reviewer Meeting:**

This was not a borderline paper, reviewers agreed that it can be improved with an additional iteration.